# Pain Neuroscience Education Reduces Pain and Improves Psychological Variables but Does Not Induce Plastic Changes Measured by Brain-Derived Neurotrophic Factor (BDNF): A Randomized Double-Blind Clinical Trial

**DOI:** 10.3390/healthcare13030269

**Published:** 2025-01-30

**Authors:** Silvia Di-Bonaventura, Aser Donado-Bermejo, Federico Montero-Cuadrado, Laura Barrero-Santiago, Lucía Pérez-Pérez, José Vicente León-Hernández, Josué Fernández-Carnero, Raúl Ferrer-Peña

**Affiliations:** 1Department of Physical Therapy, Occupational Therapy, Rehabilitation, and Physical Medicine, Rey Juan Carlos University (URJC), 28933 Alcorcón, Spain; silvia.dibonaventura@urjc.es; 2International Doctoral School, Faculty of Health Sciences, URJC, 28933 Alcorcón, Spain; aser.physio@gmail.com; 3Cognitive Neuroscience, Pain, and Rehabilitation Research Group (NECODOR), Faculty of Health Sciences, URJC, 28933 Madrid, Spain; raul.ferrer@lasallecampus.es; 4Unit for Active Coping Strategies for Pain in Primary Care, East-Valladolid Primary Care Management, Castilla and Leon Public Health System (SACYL), 47007 Valladolid, Spain; federico.montero@uva.es; 5Department of Cell Biology, Genetics, Histology, and Pharmacology, Faculty of Medicine, University of Valladolid (UVa), 47002 Valladolid, Spain; laura.barrero@uva.es; 6Nursing Department, Faculty of Nursing, UVa, 47005 Valladolid, Spain; lucia.perez@uva.es; 7Nursing Care Research Group (GICE), Faculty of Nursing, UVa, 47005 Valladolid, Spain; 8Primary Care Management Valladolid West (SACYL), 47012 Valladolid, Spain; 9Centro Superior de Estudios Universitarios La Salle (CSEU La Salle), Autonomous University of Madrid (UAM), 28049 Madrid, Spain; jv.leon@lasallecampus.es; 10Multidisciplinary Pain Research and Treatment Group, Research Excellence Group URJC-Banco Santander, 28933 Alcorcón, Spain; 11La Paz Hospital Health Research Institute, IdiPAZ, 28046 Madrid, Spain; 12Musculoskeletal Pain and Motor Control Research Group, Faculty of Sport Sciences, European University of Madrid, 28670 Villaviciosa de Odón, Spain; 13Clinical and Teaching Research Group on Rehabilitation Sciences (INDOCLIN), CSEU La Salle, UAM, 28023 Madrid, Spain

**Keywords:** pain neuroscience education, brain plasticity, BDNF, chronic pain

## Abstract

Introduction: PNE, focusing on cognitive aspects, aims to change patients’ beliefs about pain. However, it is unclear if these cognitive changes are sufficient to influence other components such as neuroplastic changes. Objective: To assess whether 3-h pain neuroscience education (PNE) can induce changes in brain-derived neurotrophic factor (BDNF) levels and pain intensity in chronic pain patients. Methods: A double-blind randomized clinical trial was conducted with 66 participants aged 18–65 years old (50.86 ± 8.61) with chronic primary musculoskeletal pain divided into two groups: an intervention group receiving 3-h PNE lecture and a control group that received an educational booklet. Primary outcomes included plasma BDNF levels and perceived pain intensity (VAS). Secondary outcomes included anxiety (HADS-A), depression (HADS-D), catastrophizing (PCS), kinesiophobia (TSK), stress (PSS), and knowledge about pain. Measurements were taken in both groups before and after a three-hour intervention. Data were analyzed using paired *t*-tests and Cohen’s d for effect sizes. Results: The results showed no significant changes in BDNF levels for the PNE lecture group (*p* = 0.708) or the educational booklet group (*p* = 0.298). Both groups showed significant reductions in pain intensity (PNE: *p* < 0.001, d = 0.70; booklet: *p* = 0.036, d = 0.39). Secondary variables, such as knowledge (PNE: *p* < 0.001, d = −0.972; booklet: *p* < 0.001, d = −0.975) and anxiety (PNE: *p* < 0.001, d = 0.70; booklet: *p* = 0.035, d = 0.39), also showed significant improvements. Conclusions: PNE did not significantly change BDNF levels but effectively improved pain intensity, pain-related knowledge, and other clinical variables. These findings suggest that while PNE has cognitive benefits, it may not be sufficient to induce immediate neurobiological changes. Further research is needed to explore long-term effects and incorporate additional therapeutic domains.

## 1. Introduction

Chronic pain, defined as pain that persists or recurs for more than 3 months [1], is estimated to affect approximately one-fifth of the world’s population [1], representing a substantial detriment to individual quality of life [2], and imposing significant economic burdens on society [3]. Chronic pain is a leading cause of disability and the costs of its management significantly exceed the combined expenditures associated with the treatment of cancer, diabetes, and cardiovascular disease [4]. In addition to its physical impact, it also affects multiple dimensions of individuals’ lives, including significant neurocognitive alterations [4], such as reductions in processing and psychomotor speed [5], along with impairments in executive functions such as attention, working memory, organization, cognitive flexibility, and inhibitory control, in addition to affecting retention and conditioned learning and extinction processes [6,7,8]. These neurocognitive alterations are underpinned by maladaptive neuroplastic changes in the brain, which, although detrimental, remain modifiable through targeted therapeutic interventions designed to promote adaptive plasticity.

Pain neuroscience education (PNE) represents a specific educational strategy aimed at managing chronic musculoskeletal pain, primarily addressing the cognitive–evaluative area [9]. PNE differs from other educational and cognitive–behavioral approaches by seeking to change the concept of pain, focusing on altering patients’ perceptions of what pain is, how it works, and the biological processes associated with it [10]. This educational model has been applied, both in master class format, as well as in the form of educational materials or handouts, which have proven to be useful for the reconceptualization of pain and the improvement of associated psychological variables [10].

Although PNE can influence cognitive aspects related to pain perception, there are indications that this intervention alone may not be sufficient to modify pain-coping behaviors effectively [11]. Previous studies, such as the one by Moseley, 2004 [12], have shown that cognitive-only interventions fail to significantly alter pain coping strategies in patients. This finding suggests the importance of adopting a multimodal therapeutic approach, which not only integrates cognitive components but also incorporates physical interventions and other bio–psycho–social elements. Such an approach could be more effective in inducing sustainable behavioral changes and improving the quality of life of patients suffering from chronic pain. Some authors propose for the clinical management of chronic pain to employ strategies designed from a salutogenic approach, empowering patients through the promotion of self-management practices, making the educational part a fundamental element of this therapeutic approach [13].

Given that the experience of pain is inherently multifactorial [14,15], it is essential that its effective management addresses all its component domains, not just the sensory and cognitive aspects. Although it has been recorded in the literature that certain interventions can modulate psychological factors, modulation of these factors in isolation may not significantly alter the entire pain experience, as pain includes purely sensory aspects [16]. Furthermore, we have not identified previous studies in the literature that directly examine the impact of PNE-based interventions on different neurotrophic agents or mediators of neuroplasticity.

In this context, the role of brain-derived neurotrophic factor (BDNF), an important neurotrophin involved in multiple signaling mechanisms of learning and memory that could play a key role in the initiation or maintenance of central nervous system hyperexcitability, stands out [17]. Previous evaluations have addressed the influence of various physical therapy interventions on this biomarker [18,19], but the impact of pain educational strategies on it remains unexplored.

This double-blind randomized controlled trial aims to evaluate whether addressing only cognitive aspects through pain neuroscience education can induce biological changes, with a primary focus on brain-derived neurotrophic factor levels and pain intensity in individuals with chronic musculoskeletal pain. Additionally, this study seeks to explore the potential impact of PNE on secondary psychosocial variables in this population.

## 2. Methods

### 2.1. Trial Design

This double-blind randomized clinical trial with an experimental design and a quantitative approach, registered under number NCT05736172 at ClinicalTrials.gov, was approved by the Research Ethics Committee with approval number 3011202228522. The study was conducted at the “Unidad de Estrategias de Afrontamiento Activo del Dolor”, adhering strictly to established protocols according to the CONSORT Statement criteria to ensure study quality and transparency. Additionally, the Tidier Checklist criteria for education were applied, ensuring the completeness and consistency of the information collected during the educational process.

### 2.2. Participants

Recruitment for the study was carried out using a consecutive non-probabilistic convenience sampling method at the same unit. Participants were directly contacted by the center’s director, F.M.C., from the center’s waiting list. An initial evaluation confirmed that they met the stipulated inclusion and exclusion criteria.

Inclusion criteria required participants to be of either the male or female sex, between 18 and 65 years old, experiencing musculoskeletal pain for at least three months, and a minimum pain level of 4 on a 10-point visual analog scale (VAS). Participants should not have received physiotherapeutic treatment for the specified pain within the last three months and must be capable of understanding and voluntarily signing the informed consent. Individuals with systemic, neurological, oncological, inflammatory diseases, psychiatric pathologies, pregnancy, or type II diabetes were excluded. Participant details are presented in Table 1.

### 2.3. Procedure

Following the previous telephone screening, patients interested in the study were escorted by an independent researcher to a separate room for an in-person confirmation of their eligibility. They were provided with detailed information about the study both verbally and in writing, and they signed the informed consent form, previously approved by the ethics committee of the university. Subsequently, sociodemographic data and psychosocial variables related to their pain condition were collected (detailed below in the section “Outcome”). These initial measurements also included the extraction of 5 mL of blood from the cubital vein, performed by a trained nurse between 8:00 and 10:00 a.m. to control for circadian effects. After completing these measurements, the patients were randomized and assigned to one of two intervention groups: the 3-h pain neurophysiology education (PNE) group or the educational booklet group. Immediately after the assigned intervention, whether it was the three-hour pain education session or the reading of the educational booklet, participants completed the questionnaires again and a second blood sample was taken to evaluate post-intervention changes. The entire process, including the initial data collection and post-intervention measurements, took approximately 45 min before and 45 min after the intervention.

### 2.4. Interventions

#### 2.4.1. Intervention Group (PNE)

Patients assigned to the intervention group participated in a pain neuroscience education (PNE) session, guided by a physiotherapist with over 5 years of experience in pain management. This session lasted 3 h, with a 5-min break each hour to prevent fatigue and maintain attention. The PNE session covered several key topics from the cognitive dimension of the pain neuroscience education protocol (POBTE), including basic pain science, the difference between acute and chronic pain, the role of the nervous system in pain perception, and how thoughts, emotions, and behaviors can influence pain. The physiotherapist used PowerPoint slides projected on a screen to present the material, incorporating interactive teaching methods such as discussions, visual aids, and practical examples to engage patients and facilitate understanding. The aim of the session was to help patients reconceptualize their pain, reduce fear-avoidance behaviors, and improve pain coping strategies through a comprehensive understanding of the neurophysiological mechanisms underlying chronic pain.

#### 2.4.2. Active Control Group (Booklet)

Patients assigned to the control group received an educational booklet containing information about pain (differences between damage and pain, pain as a subjective experience, and strategies for pain management). Patients were instructed to read the entire booklet over the 3-h duration, with a 5-min break each hour, mimicking the structure of the PNE session. They remained in the study environment for the full 3 h to control for environmental factors that could influence study outcomes. During this time, another physiotherapist was present to ensure compliance and answer any questions.

#### 2.4.3. Outcomes

Sociodemographic: Age, Marital status (Single, Married, Divorced, Widowed), Employment status (Employed, Unemployed, Retired, Temporary Disability, Student, Housekeeper, Other), Level of education (none, primary, secondary, university), Sex, Body Mass Index (BMI).

Variables Height and Weight: Both variables will be determined by the patient’s estimation to obtain the body mass index (BMI), not by physical measurement during the measurement process.

#### 2.4.4. Primary Outcomes

Perceived pain intensity: Pain intensity was measured using a 100 mm numeric rating scale (NRS), where 0 represents “no pain” and 100 represents the “worst pain imaginable.” Participants marked a point on the line that best reflected the pain they were experiencing at the time of measurement. Higher scores indicated higher levels of pain, and the administration required less than one minute [20].

Plasma BDNF levels: Blood samples were collected in EDTA-anticoagulant tubes and centrifuged at 1000× *g* for 15 min within 30 min of collection. The separated plasma was aliquoted into 2 mL eppendorfs and stored at −80 °C until analysis. For the analysis, human BDNF ELISA kits (Abbexa, Catalog No: abx150799, Cambridge, UK) were used according to the manufacturer’s protocol. The enzymatic reaction was stopped with a stop solution, and the optical density was measured at 450 nm using a FLUOstar Omega microplate reader (BMG LABTECH, Offenburg, Germany). BDNF concentrations were calculated using a standard curve generated with the standard solutions provided in the kit.

#### 2.4.5. Secondary Outcomes

Anxiety and Depression: Assessed using the validated Spanish version of the Hospital Anxiety and Depression Scale (HADS), which is divided into two subscales of 7 items each: (1) Depression (HADS-D), which measures depressive symptoms; and (2) Anxiety (HADS-A), which measures anxiety symptoms. The subscales in HADS showed internal consistency indices recommended for screening tools. The items in HADS demonstrated a positive correlation with the total score of the anxiety and depression subscales. HADS was found to perform well in assessing the symptom severity and caseness of anxiety disorders and depression in somatic, psychiatric, and primary care patients, and in the general population [21].

Catastrophism was assessed using the Spanish version of the Pain Catastrophizing Scale (PCS), which measures exaggerated negative thoughts and feelings related to pain. It has demonstrated adequate psychometric properties for evaluating this construct and a high internal consistency (Cronbach’s alpha of 0.92 (95% CI 5 0.91–0.93)) [22].

Chronic Pain: Measured with the 8-item self-report instrument, the Chronic Pain Grading Scale (CPGS), which evaluates pain intensity and pain-related disability. It has good internal consistency, with a Cronbach’s alpha of 0.74, similar to versions in other languages [23].

Tampa Scale for Kinesiophobia: A self-report questionnaire comprising 17 items, which measures fear of movement or reinjury. The internal consistency of the TSK is high, with Cronbach’s alpha coefficients ranging from 0.74 to 0.93, indicating strong reliability in people with musculoskeletal pain. Test–retest reliability is also good, with correlation coefficients ranging from 0.75 to 0.88 [24]

Perceived Stress Scale: A self-report questionnaire comprising 10 items, which measures the perception of stress and the extent to which situations in life are appraised as stressful. Each item is rated on a 5-point Likert scale ranging from 0 (never) to 4 (very often). The internal consistency of the PSS is high, with Cronbach’s alpha coefficients ranging from 0.78 to 0.91, indicating strong reliability. Test–retest reliability is also good, with correlation coefficients ranging from 0.67 to 0.85 [25].

Knowledge questionnaire on specific aspects of pain: Designed ad hoc, including 5 multi-answer test questions on the contents used in the education sessions.

### 2.5. Sample Size

The sample size was calculated using Epidat 4.1 software (Xunta de Galicia, Spain). In a pilot study conducted by the research team, it was observed that the average value of pain intensity assessed with the VAS in patients with pain of more than 3 months of evolution was 50 mm with a standard deviation of 25 mm. The size needed to estimate a clinically relevant difference, achieving a decrease of 20 mm in the VAS, assuming a standard deviation of 25 mm with a type I error (α) of 0.05, a power (1-β) of 80% in a two-tailed test, and assuming a loss rate of 25%, is 66 subjects (33 per group).

### 2.6. Randomization and Allocation Concealment

The randomization of participants was conducted using random number generation software to ensure an equitable distribution between the intervention and control groups. Assignment was performed by an independent researcher who did not participate in data collection or the administration of interventions. Subjects were assigned to one of the two groups using a pre-generated allocation list sealed in opaque envelopes to maintain blinding. These envelopes were opened only after the participant had completed informed consent and the initial assessment.

### 2.7. Blinding

Both participants and researchers responsible for measurements and data analysis were blinded to who received the experimental treatment and who received the placebo. Only the team responsible for providing the education knew the participants’ group assignments. The personnel administering the questionnaires were located in a separate room, completely unaware of the participants’ identities and their group assignments. Patients were unaware of which intervention represented the control group and which represented the experimental intervention; they were only informed that they were participating in a pain education session.

### 2.8. Statistical Methods

Several statistical analyses were performed using SPSS software version 29 to assess the effects of an educational intervention on pain neurophysiology in two groups: one receiving a lecture format intervention and the other receiving the educational booklet. Parametric tests were chosen based on the central limit theorem, which assumes a normal distribution for sample sizes above 30 subjects. Paired *t*-tests were used to compare pre- and post-intervention measures within each group. Independent *t*-tests were conducted for between-group comparisons for both independent variables (pain intensity and BDNF levels) and dependent variables (secondary and sociodemographic variables). To evaluate the efficacy of the treatment, an intention-to-treat analysis was carried out. Additionally, effect sizes were calculated using Cohen’s d to assess the magnitude of the differences observed. A value of 0.2 was referenced for a small effect size, 0.5 for a medium effect size, and 0.8 for a large effect size.

## 3. Results

Results of the intention-to-treat analysis were conducted on a total of 66 subjects divided into two study groups, each comprising 33 subjects. The PNE group consisted of 81.8% (*n* = 27) women, while the educational booklet group had 87.9% (*n* = 29) women. No statistically significant differences were found between the two study groups in the control and sociodemographic variables analyzed, guaranteeing the comparability of both study groups.

A paired sample analysis was performed to compare serum BDNF levels before and after the intervention in both groups. In the PNE group, no significant differences were found in BDNF levels before and after the intervention (*p* = 0.708; d = 0.066). Similarly, no significant differences were observed in the educational booklet group (*p* = 0.298; d = −0.19). Both groups showed statistically significant differences in pain intensity over time between pre- and post-intervention (*p* < 0.05) but not within the groups themselves. Further details related to the analysis of these variables are described in Table 2.

Results from the knowledge test administered before and after the intervention were also evaluated in both groups. In the PNE group, there was a statistically significant improvement in test outcomes (*p* < 0.001; d = −0.972). Similarly, a statistically significant improvement was observed in the educational booklet group (*p* < 0.001; d = −0.975).

Psychosocial variables were compared for both groups before and after the interventions, with both groups showing statistically significant reductions in anxiety and pain intensity. The PNE group demonstrated a medium effect size (*p* < 0.001; d = 0.70) and the educational booklet group a small effect size (*p* = 0.036; d = 0.39) for pain intensity reductions. Additional psychosocial variables and their comparisons are described in Table 3.

## 4. Discussion and Conclusions

### Discussion

The primary objective of this study was to assess whether PNE, which incorporates techniques such as educational pain analogies, re-education of patient misconceptions regarding disease pathogenesis, and guidance about lifestyle, can induce changes in BDNF levels and pain intensity in patients with chronic pain, with the secondary objective being to examine its effects on psychosocial variables associated with pain.

The impact of BDNF in the context of chronic pain pathophysiology [26] has received considerable attention, however, we have not identified previous literature directly assessing the effects of a PNE intervention on serum BDNF levels. BDNF is a neurotrophin proposed as a biomarker in the presence of neuropathic pain and chronic pain because of its association with neuroinflammatory processes [19,27,28], as well as for its association with neuroplasticity [29,30], and its contribution to long-term potentiation (LTP), being a fundamental mechanism for learning and memory processes [31]. However, the results of our study indicated that BDNF plasma levels were not statistically significantly altered after a 3-h PNE intervention.

On the other hand, previous studies have shown that even brief interventions can produce an immediate increase in serum BDNF levels. For example, it has been documented that short physical exercise sessions, as short as 20 min, can increase BDNF levels [32,33] in the immediate term. Moreover, in a study by Parchure et al. [34], continuous transcranial magnetic theta burst stimulation (cTBS) of up to 30 min in a single administration aimed to evaluate changes in cortical excitability in chronic stroke patients and possible mediation through BDNF. Their findings suggest that BDNF could play a role in neuroplastic responses to TMS.

The lack of changes in BDNF after the educational intervention in our study could be explained by several factors. First, the intervention dose chosen, although higher than the minimum necessary dose described by Suso et al. [35], might not have been sufficient to induce changes in BDNF levels in an educational context. Second, other mechanisms previously described in the literature could be mediating learning in this context, such as cholinergic systems or neurotrophic factors like NGF or neurotrophin NT3 [36]. Glutamate-dependent potentiation, particularly in the hippocampus, could also be involved here [37]. There could have been an initial glutamate-mediated potentiation rather than BDNF-mediated potentiation, and this mechanism could be sufficient to generate changes in perceived pain intensity and psychosocial variables without altering serum BDNF levels significantly in the short term. Identification of the precise mechanisms mediating learning consolidation is crucial for optimizing future educational interventions. In particular, glutamate-dependent potentiation in the hippocampus suggests that there may be alternative pathways through which learning could occur and improvement in psychosocial variables and pain perception can be achieved [37], without necessarily increasing BDNF levels, and be equally relevant in clinical contexts.

On the other hand, the results of our study indicated a statistically significant decrease in pain intensity in both groups after the intervention. This contrasts with the findings of a previous systematic review by Louw et al. [38], in which studies that applied educational approaches in isolation did not produce decreases in pain intensity scores, whereas those that combined NSP with a physical intervention did decrease pain intensity. The high heterogeneity in educational protocols could explain these differences, this heterogeneity being a common problem in this field [39].

In line with our intervention and our findings, a recent study [40] has observed that education combined with exercise may not be superior to education alone for pain reduction and improvements in function and quality of life. Other recent studies, such as that of Vicente-Mampel et al. [41] also find a statistically significant difference in pain intensity when applying an educational intervention with similar characteristics to the one used in our study. In the pragmatic trial by Galan-Martin et al. [42], a decrease in pain intensity was observed in the group that received education and exercise versus treatment as usual. Furthermore, the addition of PNE to a telerehabilitation program in patients with carpal tunnel syndrome produced improvements in symptom intensity [43], with similar results observed implementing PNE in telerehabilitation in patients with knee osteoarthritis [44].

The educational intervention employed in our study resulted in statistically significant improvements in psychological variables, such as anxiety, suggesting that both PNE in the 3 h lecture format and the educational booklet format may have a positive impact on the short-term psychological well-being of chronic pain patients.

The results of this study are in line with the existing literature on the positive effects of PNE on various psychosocial variables. In relation to anxiety and depression, Kararti et al. [45] and Lepri et al. [46] observed significant improvements in patients with fibromyalgia and other chronic musculoskeletal pain conditions. Regarding catastrophizing, the studies by Kararti et al. [45], Galan-Martin et al. [42], Lepri et al. [46], and Meeus et al. [47] reported a remarkable reduction in this variable, even in patients with different chronic pain conditions. Finally, regarding kinesiophobia, both Kararti et al., 2024 [45] and Galan-Martin et al. [42] and the intervention with PNE in telerehabilitation programs for patients with carpal tunnel syndrome [43] showed positive results.

## 5. Conclusions

The findings of our study highlight that the mechanisms underlying educational interventions in chronic pain management are multiple and do not appear to be sufficiently explored. Although no statistically significant changes in serum BDNF levels were observed after intervention, the reduction in pain intensity and improvements in psychological variables underscore the value of educational interventions as a therapeutic strategy in this study population. Recently, proposals have emerged that attempt to evolve education in pain patients [48], combining the reconceptualization of pain with other approaches not only directed to this cognitive sphere of pain, but also to emotional and behavioral aspects and the development of coping and self-management skills. Future research should focus on exploring these proposals for improvement.

## 6. Limitations

The present study has some limitations that should be considered when interpreting the obtained results. First, BDNF levels can be affected by the Val66Met genetic variant, which involves a substitution of valine for methionine at position 66 of the BDNF gene, potentially altering BDNF secretion and activity in the brain [34,49,50,51]. Previous studies have demonstrated that carriers of this variant have a reduced capacity to release BDNF in response to neuronal activity, which may negatively influence synaptic plasticity and memory processes. Although several intervention studies and animal models have shown that BDNF can change within 15 min if an adequate stimulus is provided [52,53,54], we cannot fully assure these results in the context of pain education due to the lack of specific studies measuring this effect. Therefore, it is possible that changes in BDNF levels may not be detectable within three hours in the context of pain education. Future studies should consider evaluating these effects over the long term to provide a more comprehensive understanding. Additionally, the study includes both fibromyalgia patients and those with other types of chronic pain, which could introduce variability in the results due to differences between these two populations. Another significant limitation is the use of a consecutive non-probabilistic sampling method, which might have favored the inclusion of individuals predisposed to participate in educational interventions, potentially influencing the study’s results.

## 7. Practical Implications

The generalizability of the findings from this trial is notable due to several factors. Firstly, we demonstrated that a three-hour intervention could have a significant effect on reducing pain intensity and improving various psychosocial variables. This suggests that even brief interventions may be beneficial in the short term for patients with chronic pain, which is encouraging for their application in time-constrained clinical settings. However, it is important to recognize that, although benefits are observed, this duration may not be sufficient to substantially modify pain coping behaviors, as it does not fully address the emotional and affective domains beyond the cognitive ones. To generate substantial and long-lasting changes in pain coping behaviors and consolidate the adaptations shown, it is necessary to address aspects related to the affective sphere and skill development, beyond the cognitive domain that traditional educational strategies like classical PNE primarily focus on. This involves applying principles of meaningful learning theories, adapting learning objectives according to the cognitive load required for each activity (e.g., using Bloom’s taxonomy), and incorporating strategies that consolidate adaptive behaviors.

Finally, the significant improvement in various variables observed in both groups highlights that, at a cognitive level, even a booklet can have positive short-term effects on psychological variables, underscoring the potential efficacy of simple educational interventions in pain management. It would be beneficial to replicate this study in different contexts and with diverse populations to confirm these findings and ensure that the observed effects are not limited to a specific group of patients.

## Figures and Tables

**Table 1 healthcare-13-00269-t001:** Descriptive statistics of the quantitative variables by group.

Category	Variable	PNE (Mean ± SD)	Educational Booklet (Mean ± SD)	*p*-Value (Student’s t)	Effect Size (Cohen’s d)
Sociodemographic variables	Age	51.24 ± 8.12	50.48 ± 9.11	*p* = 0.723	0.09
	Body Mass Index (BMI)	26.89 ± 5.46	25.52 ± 4.68	*p* = 0.278	0.27
Pain-related variables	Number of painful areas	11.55 ± 9.72	13.73 ± 9.61	*p* = 0.363	−0.23
	Pain Evolution Time (months)	208.85 ± 130.61	199.15 ± 89.83	*p* = 0.727	0.09
	Pain Intensity (VAS)	7.87 ± 1.13	7.58 ± 1.18	*p* = 0.151	0.26
Biological variables	BDNF levels	2.02 ± 0.47	1.98 ± 0.40	*p* = 0.672	0.10
Psychosocial variables	Anxiety (HADS-A)	12.88 ± 4.05	13.09 ± 4.43	*p* = 0.840	−0.05
	Catastrophism (PCS)	31.7 ± 11.75	31.94 ± 10.18	*p* = 0.929	−0.02
	Depression (HADS-D)	10.82 ± 4.36	10.76 ± 4.67	*p* = 0.957	0.01
	Kinesiophobia (TSK)	32.76 ± 6.85	29.09 ± 6.63	*p* = 0.031	0.54 *

* *p* < 0.05.

**Table 2 healthcare-13-00269-t002:** Descriptive statistics of the categorical variables.

Variable	Category	PNE	Educational Booklet	*p*-Value
		(*n* = 33)	(*n* = 33)	(Chi-Square χ^2^)
Sex	Men	6 (18.2%)	4 (12.1%)	*p* = 0.492
	Women	27 (81.8%)	29 (87.9%)	
Marital Status	Single	4 (12.1%)	6 (18.2%)	*p* = 0.636
	Married	23 (69.7%)	18 (54.5%)	
	Widowed	5 (15.2%)	8 (24.2%)	
	Divorced	1 (3.0%)	1 (3.0%)	
Work Status	Full time work	11 (33.3%)	14 (42.4%)	*p* = 0.696
	Part time work	8 (24.2%)	6 (18.2%)	
	Unemployed	5 (15.2%)	2 (6.1%)	
	Student	3 (9.1%)	3 (9.1%)	
	Retired	6 (18.2%)	8 (24.2%)	
Study Level	No Study	1 (3.0%)	0 (0%)	*p* = 0.465
	Primary	8 (24.2%)	9 (27.3%)	
	Secondary	14 (42.4%)	18 (54.5%)	
	University	10 (30.3%)	6 (18.2%)	

Significance level: *p* < 0.05.

**Table 3 healthcare-13-00269-t003:** Comparison between groups.

	Variable	Group	PRE	POST	Mean Differences	*p*-Value (Student’s t); Cohen’s d
Psychosocial	Anxiety (HADS-A)	PNE	12.88 ± 4.05	11.32 ± 4.09	1.56 ± 1.24	*p* = < 0.001; d = 0.70 *
		Educational Booklet	13.09 ± 4.43	11.42 ± 4.37	1.67 ± 1.25	*p* = < 0.001; d = 0.70 *
			−0.21 ± 0.38	−0.10 ± 0.48		
		*p*-value; Cohen’s d	*p* = 0.840; d = −0.05	*p* = 0.928; d = −0.23		

	Depression (HADS-D)	PNE	10.82 ± 4.36	9.61 ± 3.87	1.21 ± 1.24	*p* = 0.014; d = 0.47 *
		Educational Booklet	10.76 ± 4.67	10.16 ± 4.20	0.60 ± 1.25	*p* = 0.269; d = 0.20
			0.06 ± 0.31	−0.55 ± 0.33		
		*p*-value; Cohen’s d	*p* = 0.957; d = 0.01	*p* = 0.595; d = −0.13		

	Catastrophism (PCS)	PNE	31.7 ± 11.75	29.58 ± 12.55	2.12 ± 1.98	*p* = 0.058; d = 0.35
		Educational Booklet	31.94 ± 10.18	27.71 ± 10.70	4.23 ± 1.92	*p* = < 0.001; d = 0.87 *
			−0.24 ± 1.58	−1.87 ± 1.85		
		*p*-value; Cohen’s d	*p* = 0.929; d = −0.02	*p* = 0.530; d = 0.16		

	Kinesiophobia (TSK)	PNE	32.76 ± 6.85	30.10 ± 8.07	2.66 ± 1.22	*p* = 0.002; d = 0.61 *
		Educational Booklet	29.09 ± 6.63	27.45 ± 6.71	1.64 ± 1.24	*p* = 0.051; d = 0.36
			3.67 ± 0.22	2.65 ± 1.36		
		*p*-value; Cohen’s d	*p* = 0.015; d = 0.54 *	*p* = 0.166; d = 0.35		

	Percieved Stress (PSS)	PNE	34.58 ± 9.31	33.97 ± 9.67	0.61 ± 1.21	*p* = 0.251; d = 0.21
		Educational Booklet	33.12 ± 8.73	32.71 ± 8.00	0.41 ± 1.19	*p* = 0.581; d = 0.10
			1.46 ± 0.58	1.26 ± 1.67		
		*p*-value; Cohen’s d	*p* = 0.257; d = 0.16	*p* = 0.289; d = 0.14		

	Pain Intensity (VAS)	PNE	7.87 ± 1.13	7.72 ± 1.20	0.15 ± 0.15	*p* = 0.036; d = 0.39 *
		Educational Booklet	7.58 ± 1.18	7.18 ± 1.28	0.40 ± 0.16	*p* = 0.035; d = 0.39 *
			0.29 ± 0.05	0.54 ± 0.08		
		*p*-value; Cohen’s d	*p* = 0.151; d = 0.25	*p* = 0.089; d = 0.43		

Biochemical	Plasma BDNF Levels	PNE	2.02 ± 0.47	2.00 ± 0.46	0.02 ± 0.34	*p* = 0.708; d = 0.66
		Educational Booklet	1.98 ± 0.40	2.07 ± 0.46	−0.88 ± 0.46	*p* = 0.298; d = 0.19
			0.04 ± 0.10	−0.07 ± 0.11		
		*p*-value; Cohen’s d	*p* = 0.672; d = 0.43	*p* = 0.525; d = 0.46		

* *p* < 0.05.

## Data Availability

The data used in this study are available upon request. To access the dataset, interested parties must justify the need for its use and contact the corresponding author. Access will be subject to ethical and regulatory considerations and will be granted only for legitimate research purposes.

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
