# Peer review of "Pain Neuroscience Education Reduces Pain and Improves Psychological Variables but Does Not Induce Plastic Changes Measured by Brain-Derived Neurotrophic Factor (BDNF): A Randomized Double-Blind Clinical Trial"

_healthcare, 2025, doi:10.3390/healthcare13030269_

Round 1

Reviewer 1 Report

Comments and Suggestions for Authors

Title

I suggest removing "3h" from the title because it is very specific information, which can be mentioned in the abstract and in the method of the manuscript.

Abstract

I suggest starting with the Introduction, then the objective.

In the method it should be missing to mention: clinical conditions of the participants, healthy people or with acute or chronic pain, mean and standard deviation of age, the instruments used, it is not to mention the variables.

The p-statistic is written in lowercase.

Introduction

I would suggest that you improve the introduction. The Introduction talks about Pain Neuroscience Education, but it does not talk about synaptic plasticity, does not describe extensively what is a brain-derived neurotrophic factor, and studies with the topic, and it does not define chronic pain. The Introduction should end by mentioning the general objective, the specific objectives, and the hypotheses. The study design should be the first section of the method.

Method

Study design: should indicate whether it is "a randomized clinical trial with an experimental design and a quantitative approach".

It is randomized if they randomly assign participants to groups.

It is experimental if they manipulated the independent variables.

It is quantitative if it is based on the collection and analysis of numerical data.

Participants: The inclusion criteria fail to mention whether they include both sexes. Table 1, they do not specify how many people are women. In Table 1, specify the statistical test you used to find significant differences. Define the socio-demographic characteristics of the participants (number of participants, sex, age, academic level, and other relevant related characteristics). It is preferable to present this information in a table, divided into groups that allow you to compare these characteristics. You should specify whether the groups were matched. If not, this should be mentioned in the limitations.

Instruments and Materials: describe the questionnaires, the psychological tests used (defining what they measure, how they are assessed and mentioning the psychometric properties, which allows to verify that the tests were adequate). Describe in detail the tasks and equipment used.

Data analysis: describe the analyses used, indicating which are the independent and dependent variables. Justify why T-Test was used.

Procedures: Describe in detail how the study was conducted so that any researcher can replicate the study, starting with mentioning that it was approved by the Ethics Committee, as mentioned by the authors.

Results

I suggest that table 2 be shown in the participants section, as they are not the actual results of the study's objective, they are the characteristics of the participants you recruited in order to conduct the research. In table 2 you should report the statistical test values.

Table 3 should show the results of the statistical test used.

In table 3 you show the results of the BDNF plasma. Can this be considered a psychosocial variable?

Practical Implications

The authors suggest on lines 374-379 "This suggests that even brief interventions can be beneficial in the short term for patients with chronic pain, which is encouraging for their application in clinical settings with time constraints. However, it is important to acknowledge that while benefits are observed, this duration may not be sufficient to substantially modify pain coping behaviors, as it does not fully address emotional and affective domains beyond the cognitive ones." It remains to be suggested what would need to be implemented to generate substantial changes in pain-coping behavior and generate plastic changes.

Author Response

Dear reviewer,

Thank you for the insightful feedback and suggestions, which have significantly contributed to refining our manuscript. We trust that we have addressed your comments effectively and are available for any additional clarifications or modifications.

  1. Title

I suggest removing "3h" from the title because it is very specific information, which can be mentioned in the abstract and in the method of the manuscript.

Response: Thank you for your feedback, it’s very appropriate and we have changed the title and included this detail in the abstract and in the methods section of the manuscript.

  1. Abstract

I suggest starting with the Introduction, then the objective.

In the method it should be missing to mention: clinical conditions of the participants, healthy people or with acute or chronic pain, mean and standard deviation of age, the instruments used, it is not to mention the variables.

The p-statistic is written in lowercase.

Response: Thank you very much for the contributions in this section. We have applied all of them in each section of the abstract. You can find it underlined in yellow to facilitate the revision.

  1. Introduction

I would suggest that you improve the introduction. The Introduction talks about Pain Neuroscience Education, but it does not talk about synaptic plasticity, does not describe extensively what is a brain-derived neurotrophic factor, and studies with the topic, and it does not define chronic pain. The Introduction should end by mentioning the general objective, the specific objectives, and the hypotheses. The study design should be the first section of the method.

Response: Your suggestion is entirely valid. We have incorporated it by providing a more comprehensive explanation of synaptic plasticity, offering a clearer definition of BDNF, and introducing chronic pain with its appropriate definition. Additionally, we have reorganized the final section of the introduction in alignment with the proposed structure.

  1. Method

Study design: should indicate whether it is "a randomized clinical trial with an experimental design and a quantitative approach".

It is randomized if they randomly assign participants to groups.

It is experimental if they manipulated the independent variables.

It is quantitative if it is based on the collection and analysis of numerical data.

Response: We confirm that it is a simple randomized clinical trial because patients were assigned to groups with a random number generator, together with the quantitative and experimental approach. We appreciate this detail to make the manuscript more rigorous and of course we have modified it in the text to make it clear.

  1. Participants: The inclusion criteria fail to mention whether they include both sexes. Table 1, they do not specify how many people are women. In Table 1, specify the statistical test you used to find significant differences. Define the socio-demographic characteristics of the participants (number of participants, sex, age, academic level, and other relevant related characteristics). It is preferable to present this information in a table, divided into groups that allow you to compare these characteristics. You should specify whether the groups were matched. If not, this should be mentioned in the limitations.

Response: Thank you very much for the comment regarding this.

We corrected the inclusion criteria. The statistic used to obtain the p-value for each variable has been added to Table 1. The number of women corresponding to the study variables specified in Table 1 has been added to the text in the results section, and this data has been highlighted in Table 2. The sociodemographic study variables have been better described in methods, and the results of the sociodemographic variables have been divided into the intervention groups in order to be able to compare between them if there are differences prior to the analysis, and it has been highlighted in the text that no statistically significant differences were observed in practically all the variables between the two groups before the interventions, which ensures the comparability of both study groups.

  1. Instruments and Materials: describe the questionnaires, the psychological tests used (defining what they measure, how they are assessed and mentioning the psychometric properties, which allows to verify that the tests were adequate). Describe in detail the tasks and equipment used.

Response: Thank you very much for the suggestion. All questionnaires have been defined in more detail, including the required specifications, as provided in their respective validation papers. If you find any specific information missing or would like further clarification, please let us know, and we will be happy to address it.

  1. Data analysis: describe the analyses used, indicating which are the independent and dependent variables. Justify why T-Test was used.

Response: Thank you for the appreciation, in this case we have added the explanation of the use of Student's t-tests for both dependent and independent variables, and that these tests were selected by virtue of the central limit theorem for samples larger than 30 subjects per group. You can find it in the Statistical methods section.

  1. Procedures: Describe in detail how the study was conducted so that any researcher can replicate the study, starting with mentioning that it was approved by the Ethics Committee, as mentioned by the authors.

Response: Thank you very much for trying to clarify this important section. Is there anything in particular in the procedure section that has not been detailed as I expected (timing of the study, sample collection,...)? Still, in the meantime we have improved the explanation of the process to see if it is now clearer and the Ethics Committee too. You can find it now on the page X.

  1. Results

I suggest that table 2 be shown in the participants section, as they are not the actual results of the study's objective, they are the characteristics of the participants you recruited in order to conduct the research. In table 2 you should report the statistical test values.

Table 3 should show the results of the statistical test used.

Response:

Thank you for your contribution, in relation to table 1 it is considered a result in this study since it shows the comparability of the sample through the comparative statistical analysis between groups at the beginning of the study, and provides the mean and standard deviation data that describe the study sample, although it is not a specific objective of this study to describe the variables analyzed, we consider it relevant to report this result in this section to facilitate the understanding of the comparability of the subjects at the beginning of the interventions. Descriptions of the values of the statistical tests used have been added in Table 2 and Table 3.

The authors first describe sociodemographic variables, although in table 1 variables such as age or BMI do not appear at the beginning

The description in methods of the sociodemographic variables of studies showing their results has been added in Table 1.

  1. In table 3 you show the results of the BDNF plasma. Can this be considered a psychosocial variable?

Response:  Thank you for your question. BDNF plasma levels are not considered a psychosocial variable; they are primarily a biological marker reflecting neurophysiological processes such as brain plasticity, neuronal survival, and synaptic modulation. For this reason, in Table 3, we have decided to include subcategories clearly distinguishing biological and psychosocial variables to ensure clarity and avoid misinterpretation.

  1. Practical Implications

The authors suggest on lines 374-379 "This suggests that even brief interventions can be beneficial in the short term for patients with chronic pain, which is encouraging for their application in clinical settings with time constraints. However, it is important to acknowledge that while benefits are observed, this duration may not be sufficient to substantially modify pain coping behaviors, as it does not fully address emotional and affective domains beyond the cognitive ones." It remains to be suggested what would need to be implemented to generate substantial changes in pain-coping behavior and generate plastic changes.

Response:

Thank you very much for taking this into account, it is an important detail. We have specified it more explicitly in the text after that sentence.

Reviewer 2 Report

Comments and Suggestions for Authors

Dear authors,

After reviewing your study, there are certain aspects that I recommend the authors to consider. The manuscript has a misleading title. The authors are not analysing the effect on duration of the educational programme in this manuscript, nor are they doing so in a double-blind design. Please reconsider the title.

After careful reading, I believe that what they are comparing are two forms of delivery of pain-related educational content 

Similarly, the title indicates that it does not generate plastic changes. Does the existence of plastic changes only depend on changes in brain-derived neurotrophic factor (BDNF) levels?

The authors first describe sociodemographic variables, although in table 1 variables such as age or BMI do not appear at the beginning. Please review

Given the large number of variables they analyse, they could possibly group them into sections (pain-related variables, psychosocial variables, degree of knowledge etc...).

What results were achieved on the degree of knowledge gained? It is good that the participants of the study do not present differences in the level of education between groups, as this is a study in which this aspect is relevant, although the authors do not indicate the diagnostic label of the participants. They speak of persistent musculoskeletal pain, with at least 4/10 on the VAS scale. Why?

Page 3 191-121: ‘Individuals with systemic, neurological, oncological, inflammatory diseases, psychiatric pathologies, pregnancy, or type II diabetes were excluded’. 

I have a question regarding the clinical entity of the individuals in this study. Persistent musculoskeletal pain is not a systemic disease? The exclusion criteria need to be more precise.

The authors make no reference either in the results or in the discussion to the variable degree of knowledge of acquired pain. Please include this information.

Finally, it is missing in the discussion section, that the authors discuss how the contents, and not only the objectives, are delivered in each of the phases of the POBTE protocol.

Author Response

Dear reviewer,

We do appreciate the comments and suggestions provided, which have helped improve our manuscript. We hope to have addressed all the observations adequately and remain available for any further clarification or adjustments.

Dear authors,

After reviewing your study, there are certain aspects that I recommend the authors to consider.

  1. The manuscript has a misleading title. The authors are not analysing the effect on duration of the educational programme in this manuscript, nor are they doing so in a double-blind design. Please reconsider the title.

Response: Thank you for your comment. We have revised the title to make it clearer and more accurately reflect the content of the manuscript. We hope that the new title aligns better with your expectations and the focus of the study.

  1. After careful reading, I believe that what they are comparing are two forms of delivery of pain-related educational content.

Response: The choice of a booklet as the control intervention is supported by its focus on general health education, while the PNE session provides specific pain education addressing cognitive aspects typical of the condition. This distinction allows us to maintain blinding by informing participants they will receive education, with the difference lying in the content and depth of the interventions. Also, this is supported by previous studies that have utilized written materials as educational controls to minimize the influence of personal interaction and intervention bias (e.g., Ersek et al., 2008; Gallagher et al., 2013; Rantonen et al. 2013; de Rezende et al., 2016; Saper et al., 2017; Dear et al., 2017; Greeling et al., 2023).

  1. Similarly, the title indicates that it does not generate plastic changes. Does the existence of plastic changes only depend on changes in brain-derived neurotrophic factor (BDNF) levels?

Response: Thank you for your insightful question. While brain-derived neurotrophic factor (BDNF) is an important biomarker associated with neuroplasticity, it is not the sole determinant of plastic changes. To address this point, we have clarified the title to specify that the study focuses on plastic changes as measured by BDNF. This refinement aligns with your recommendation and ensures a more precise representation of the study's scope.

  1. The authors first describe sociodemographic variables, although in table 1 variables such as age or BMI do not appear at the beginning. Please review

Given the large number of variables they analyse, they could possibly group them into sections (pain-related variables, psychosocial variables, degree of knowledge etc...).

Response: Thank you for your observation. We have reviewed and updated the presentation of variables in Table 1, ensuring that key sociodemographic variables such as age and BMI appear at the beginning. Additionally, we have grouped the variables into sections to enhance clarity and improve the comprehensibility of the analysis.

  1. What results were achieved on the degree of knowledge gained? It is good that the participants of the study do not present differences in the level of education between groups, as this is a study in which this aspect is relevant, although the authors do not indicate the diagnostic label of the participants.

Response: We appreciate your observation, although we are not entirely sure if we fully understood the question. Knowledge acquisition in our study was assessed using a specific ad hoc questionnaire designed to measure the key concepts addressed in both the PNE session and the booklet. The results, which show significant improvements in knowledge in both groups (p < 0.001; d = -0.972 for the PNE group and d = -0.975 for the booklet group), have been further clarified and specified in the results section of the manuscript. If there are additional aspects you would like us to address, please let us know.

  1. They speak of persistent musculoskeletal pain, with at least 4/10 on the VAS scale. Why?

Response: We selected the criterion of persistent musculoskeletal pain with a minimum score of 4/10 on the VAS scale to ensure the inclusion of participants experiencing clinically relevant pain, the choice aligns with commonly used cutoffs in pain research to distinguish between mild and moderate pain severity (Boonstra et al. 2014), which is associated with meaningful functional and psychological impact. This approach allowed us to capture a population representative of real-world clinical practice, ensuring that the intervention was tested in individuals for whom pain education is most likely to provide therapeutic benefit.

  1. Page 3 191-121: ‘Individuals with systemic, neurological, oncological, inflammatory diseases, psychiatric pathologies, pregnancy, or type II diabetes were excluded’. 

I have a question regarding the clinical entity of the individuals in this study. Persistent musculoskeletal pain is not a systemic disease? The exclusion criteria need to be more precise.

Response: Thank you for your observation. Persistent musculoskeletal pain is not classified as a systemic disease under the ICD-11. The ICD-11 defines chronic pain, including persistent musculoskeletal pain, as a distinct clinical entity characterized by pain that persists for at least three months and is not better explained by another condition. This classification explicitly distinguishes it from systemic diseases, which involve widespread pathological processes affecting multiple organ systems, such as rheumatoid arthritis or systemic lupus erythematosus.

The exclusion criteria in our study were designed to avoid potential confounding factors. Systemic, neurological, oncological, and inflammatory diseases were excluded because they could present with pain as a symptom, potentially confounding the evaluation of pain mechanisms specific to chronic primary pain. Additionally, these conditions are known to influence levels of brain-derived neurotrophic factor (BDNF), which could have introduced variability in our biomarker measurements. By excluding these diseases, we aimed to ensure the validity and reliability of our findings.

  1. The authors make no reference either in the results or in the discussion to the variable degree of knowledge of acquired pain. Please include this information.

Response: Thank you very much for this point, the information has been included in page 12.

  1. Finally, it is missing in the discussion section, that the authors discuss how the contents, and not only the objectives, are delivered in each of the phases of the POBTE protocol.

Response: We appreciate this comment and agree on the importance of detailing how the content was delivered in this specific study. While the POBTE framework includes cognitive, emotional, and behavioral strategies, this study focused exclusively on the cognitive dimension of the protocol. The educational session aimed to facilitate conceptual understanding of pain by using techniques such as practical examples, visual aids, and interactive discussions. These methods were chosen to enhance engagement and promote the reconceptualization of pain. The cognitive focus was intentional, as the single-session design limits the implementation of more complex or longitudinal strategies inherent to the full POBTE protocol. We included this clarification in the discussion to better contextualize this study.